# Understanding NAFLD: From Case Identification to Interventions, Outcomes, and Future Perspectives

**DOI:** 10.3390/nu15030687

**Published:** 2023-01-29

**Authors:** Daniel Clayton-Chubb, William Kemp, Ammar Majeed, John S. Lubel, Alex Hodge, Stuart K. Roberts

**Affiliations:** 1Department of Gastroenterology, The Alfred Hospital, Melbourne, VIC 3004, Australia; 2Central Clinical School, Monash University, Melbourne, VIC 3004, Australia; 3Department of Gastroenterology, Eastern Health, Box Hill, VIC 3128, Australia; 4Eastern Health Clinical School, Monash University, Box Hill, VIC 3128, Australia

**Keywords:** NAFLD, MAFLD, NASH, non-invasive tests, metabolic syndrome, cirrhosis, cardiovascular disease

## Abstract

While non-alcoholic fatty liver disease (NAFLD) is a prevalent and frequent cause of liver-related morbidity and mortality, it is also strongly associated with cardiovascular disease-related morbidity and mortality, likely driven by its associations with insulin resistance and other manifestations of metabolic dysregulation. However, few satisfactory pharmacological treatments are available for NAFLD due in part to its complex pathophysiology, and challenges remain in stratifying individual patient’s risk for liver and cardiovascular disease related outcomes. In this review, we describe the development and progression of NAFLD, including its pathophysiology and outcomes. We also describe different tools for identifying patients with NAFLD who are most at risk of liver-related and cardiovascular-related complications, as well as current and emerging treatment options, and future directions for research.

## 1. Introduction

### History, Prevalence, Nomenclature Controversies

Liver disease is the 11th most common cause of mortality and the 15th most common cause of morbidity globally [1]. While liver disease has many aetiologies including viral hepatitis, alcohol, autoimmune and genetic disorders, the proportion attributable to non-alcoholic fatty liver disease (NAFLD) has been growing rapidly—paralleling the global obesity epidemic (Figure 1). Patients with NAFLD have increased mortality compared with matched control populations due to cardiovascular, liver and/or malignant causes [2]. 

NAFLD encapsulates a spectrum of disease, from simple hepatic steatosis (defined as >5% of hepatocytes containing fat) through to non-alcoholic steatohepatitis (NASH), fibrosis, and cirrhosis [4]. The diagnostic criteria for NAFLD rely on ‘ruling-out’ hepatic steatosis due to other factors such as alcohol and drugs. Indeed, as the name suggests, alcohol is explicitly listed as an exclusion criterion. Since NAFLD was first coined in 1987 [5], its incidence has risen dramatically, with an estimated 25% of the world’s adult population now affected [6,7]. Furthermore, NAFLD is on track to become the leading cause of liver-related morbidity, mortality, and transplant [8,9,10]. 

Defining a disease entity via predominantly negative criteria has attracted criticism. As early as 2002, questions were raised regarding NAFLD’s nomenclature and aetiology following research associating the condition with metabolic abnormalities [11] and the demonstration of a link between the pathophysiology of NAFLD with insulin resistance and the metabolic syndrome [12,13]. Additionally, new evidence emerged that NAFLD may worsen outcomes in patients with chronic hepatitis C [14]—however, patients, by definition, were not classified as having concomitant NAFLD. Multiple position statements and editorials over the next 18 years espoused the merits of renaming NAFLD to better encapsulate the pathogenesis of the disease [15,16,17]. 

In this context, the term metabolic dysfunction associated fatty liver disease (MAFLD) was recently coined in 2020 by a large international consensus group [18,19] that utilises ‘inclusion-based diagnostic criteria’ of having hepatic steatosis (determined either by imaging, blood biomarkers or scores, or histology) with one or more of: overweight, type 2 diabetes mellitus (T2DM), or two or more features of metabolic dysfunction [18]. Although there are several favourable outcomes associated with renaming NAFLD to MAFLD including a stronger link between the underlying pathophysiology and disease state that relates diagnostic criteria to phenotype and pathological outcomes, as well as allowing multi-aetiologic liver disease diagnoses that may include NAFLD, the proposed nomenclature change is not without controversy [20]. Several concerns have been raised, including: (i) while NAFLD is an imperfect name, changing nomenclature would be better undertaken once identifiable pathophysiological subtypes are identified; (ii) the concept and definition of metabolic health or abnormalities are contentious; (iii) various regulatory agencies use the histological presence of NASH as a key criteria for drug trials, and MAFLD implies removing NASH as a concept; (iv) the implications for patient and non-specialist disease recognition are unknown; and (v) there was a perceived lack of engagement with non-physician interest groups to coalesce around a new nomenclature that better serves all relevant parties. Given this debate is on-going and the revised nomenclature is relatively recent, our review will focus on NAFLD, highlighting developments in the field and opportunities for improving outcomes. 

## 2. Body

### 2.1. The Pathogenesis of NAFLD and NASH

The pathogenesis of NAFLD remains incompletely understood, and likely varies between individuals. However, the “multiple-hit hypothesis” is a useful framework to aid in our understanding of this condition [21,22]. The development of hepatic steatosis is the “first hit”, and sequential or simultaneous combinations of genetic susceptibilities, dietary habits, lifestyle factors, metabolic dysfunction (primarily insulin resistance), and/or alterations in the gut microbiome are all potential contributors to the development of more advanced liver disease [23,24]. Through this pathway, the subsequent or concurrent activation of the adaptive and innate immune system leads to the activation of hepatic stellate cells and Kupffer cells, promoting fibrosis and cirrhosis [25].

### 2.2. Hepatic Fatty Acid Metabolism

The liver is the major site of fatty acid metabolism in humans [26]. Fatty acids—taken into hepatocytes via plasma or formed by de novo lipogenesis (DNL)—are primarily used for energy production through oxidation or are excreted back into plasma in the form of very low density lipoproteins (VLDL) [26,27]. When metabolically healthy, the esterification of free fatty acids (FFAs) to glycerol (forming triglycerides) and subsequent storage in hepatocytes is rare; any significant hepatic accumulation of fat is, thus, considered abnormal. In obese NAFLD, there are three primary sources of hepatic fatty acid accumulation: DNL (15–40%) [28], dietary intake (~15%) [29], and (predominantly adipose tissue-derived) serum esterified fatty acids (~59%) [29]. While the rate of DNL is significantly different between NAFLD and non-NAFLD subjects [30,31], NAFLD is also characterised by increased adipose tissue fatty acid flux and peripheral insulin resistance [31]. Indeed, both adipose tissue insulin resistance [32] and the anatomic location of fat [33] (visceral vs. subcutaneous) may influence the development of NAFLD. As discussed in recent reviews of adipose tissue and insulin resistance [34,35] as well as via recent lipidomic studies [36], it is likely that our understanding of the role of extra-hepatic adipose tissue and fatty acid flux on NAFLD development will continue to expand. 

However, irrespective of their source, excess hepatic fatty acids and their metabolic products may be toxic. Fatty acids—either via incomplete oxidation or pathologic metabolic pathways—may lead to the development of sphingolipids (such as ceramides) and/or diacylglycerols which can worsen insulin resistance [37,38,39]. Further, in the presence of an excess of fatty acids, a subgroup of patients are vulnerable to direct cellular insults. FFAs may induce the enzymatic production of reactive oxygen species (ROS) which cause cellular damage and hepatotoxicity [40]. Dysfunctional mitochondria, when exposed to FFAs, may also precipitate the generation of ROS—though whether mitochondria are pathogenic or bystanders in the NAFLD/NASH process is an area of active research [41]. ROS generation is a likely contributor to NASH and to the development of progressive fibrosis and cirrhosis. An improved understanding of these processes may also lead to new treatment options as dysfunctional metabolic pathways may directly contribute to hepatic steatogenesis. For example, acetyl-CoA carboxylase is required for DNL. As mentioned above, DNL is a significant contributor to hepatic steatosis; given this, pharmacotherapy inhibiting acetyl-CoA carboxylase has been studied in rodents [42,43] and humans [44] and may reduce both hepatic steatosis and fibrosis. Future work will likely elucidate additional pathophysiological pathways and potentially add to our treatment armamentarium. 

### 2.3. Genetics and Heritability

Multiple studies have found that NAFLD is partly heritable. Several gene variants have been implicated in the presence, pathogenesis, and severity of NAFLD [45,46], though only 10–20% of NAFLD cases are associated with currently recognised gene variants [47]. The most studied and well recognised of these is *PNPLA3*, which was first linked to NAFLD by a 2008 genome-wide association study [48]. *PNPLA3* encodes a protein on hepatocyte lipid droplets. When dysfunctional, it may accumulate, impairing the access of other lipase enzymes to the lipid droplets, leading to impaired triglyceride hydrolysis and consequently lipid accumulation in hepatocytes [27]. Other genetic abnormalities affecting lipid transport and metabolism are implicated in the pathogenesis of NAFLD, including *TM6SF2*, *GCKR*, and *HSD17B13* [46]. When pathological, these single nucleotide polymorphisms (SNPs) may influence VLDL transport, FFA production and hepatocellular uptake, and glycolysis. In essence, SNPs affecting almost all stages of hepatic fatty acid metabolism are implicated in the development of NAFLD, emphasising that many metabolic pathways are involved in hepatic steatogenesis. Additionally, well characterised metabolic disorders such as familial hypobetalipoproteinaemia may cause a heritable form of hepatic steatosis [49]. While familial hypobetalipoproteinaemia may not be related to insulin resistance and is commonly associated with low serum lipids [49] and possible cardioprotection [50], its impact on histological liver disease severity and progression remains uncertain [51]. 

### 2.4. Dietary Intake

Multiple dietary habits contribute to the development of NAFLD and can be considered in terms of their overall energy load and/or energy-containing macronutrient constituents which may have differing biological effects. Both observational and interventional studies have shown that excess caloric intake is associated with NAFLD irrespective of diet composition [52,53]. Additionally, a study of NAFLD pre- and post-dietary advice showed an association between weight gain and the development of NAFLD as well as weight loss with NAFLD resolution [52]. Mechanistically, increased dietary intake of carbohydrate (increasing hepatic substrates for DNL) or FFAs may overwhelm the capacity to manage FFA hepatocyte flux, promoting NAFLD development. Macronutrient constituents also matter. For example, glucose and fructose—which are energetically equivalent—influence metabolic pathways and affect steatogenesis differently [54,55]. Similarly, saturated fatty acids (SFAs) have different biological effects than mono- or poly-unsaturated fatty acids (MUFAs or PUFAs) [56]. In one over-feeding trial comparing SFA intake to unsaturated fatty acid and simple carbohydrate intake, SFA-based overfeeding was associated with significantly higher hepatic fat content than other overfeeding—though all participants experienced worsened hepatic steatosis [57]. Additional experimental studies have shown that even amongst unsaturated fatty acids there are differences, for example, omega-3 PUFAs have been shown to reduce liver fat content when compared with olive oil [58], possibly via the suppression of hepatic DNL and concurrently increased fatty acid oxidation. Cross-sectional observational studies also show that macronutrient composition is important independent of total caloric intake. Inverse associations between PUFA intake and NAFLD have been shown [59], as have direct links between sugar-sweetened beverages and the presence of NAFLD [60,61]. Specific foodstuffs may also play a role in the presence of NAFLD. One study has shown an association between red and/or processed meat and NAFLD [62], and there are possible inverse associations between NAFLD and green leafy vegetable intake [63]. There are additional micronutrient dietary factors that may influence the development of NAFLD and NASH—these are listed in Table 1. For more comprehensive reviews, see Pickett-Blakely et al. [64] and Licata et al. [65].

Other non-energetic dietary constituents have benefits in NAFLD and liver disease. For example, coffee intake has been previously inversely linked to cardiovascular disease, cancer incidence, and all-cause mortality [83,84]. Its use is inversely associated with abnormal liver function tests (LFTs) and liver fibrosis in various diseases including hepatitis B, hepatitis C, and NAFLD [85,86]. It is associated with a reduced risk of developing hepatocellular carcinoma (HCC) [87], a leading cause of mortality in liver disease. Multiple mechanisms have been proposed [88], including reducing the deposition of liver fat and collagen and increasing glutathione synthesis (ameliorating damage from ROS). These mechanisms are not just due to caffeine—coffee contains multiple bioactive compounds, including chlorogenic acid and other polyphenols, and decaffeinated coffee has some benefit in human HCC [87] and murine NAFLD [89]. Supporting this, a U.K. study has shown similar benefits of caffeinated compared with decaffeinated coffee consumption on outcomes in chronic liver disease (CLD) [90]. Further, considering caffeine in isolation, total intake is not associated with NAFLD prevalence nor degree of fibrosis [91]. 

### 2.5. Physical Activity

In addition to diet, other lifestyle factors are implicated in the development of NAFLD. Multiple observational studies have shown an association between a sedentary lifestyle and the incidence and prevalence of NAFLD [92,93,94]. However, the amount of physical activity required to reduce the incidence of NAFLD is relatively high. In one study, when adjusting for covariates (including visceral adiposity and insulin resistance) only those in the highest quartile of metabolic equivalents had a reduced hazard of developing NAFLD [95]. Indeed, even short-term periods of inactivity can worsen metabolic health—one study showed 14 days of marked inactivity caused significant deterioration in insulin sensitivity and increased liver fat [96]. Recent meta-analyses of interventional studies have also shown that both aerobic and resistance training can improve hepatic steatosis [97,98]—often independent of weight change. However, in addition to structured exercise, non-exercise physical activity is an important marker of health. It has been shown to be independently inversely associated with cardiovascular disease risk and all-cause mortality [99,100]. While this area is under-studied in NAFLD, one study has shown that patients with NAFLD spend more time being sedentary and walk fewer daily steps than controls [92]. In addition to the associations between an active lifestyle and weight control, exercise may independently improve insulin sensitivity [101] (improving hepatic energy kinetics) and increase hepatic VLDL clearance [102]. 

### 2.6. Metabolic Dysfunction and Insulin Resistance 

Metabolic dysfunction and insulin resistance are strongly associated with NAFLD; indeed, insulin resistance is often described as a key pathogenic feature of NAFLD. However, it is important to note that insulin resistance in NAFLD is not only present within the liver itself (leading to increased hepatic DNL), but also in skeletal muscle (impairing glucose disposal) and adipose tissue (causing lipolysis and increased fatty acid delivery to the liver) [28,103]. Furthermore, the relationship between NAFLD and insulin resistance is not unidirectional. Various data link hepatokines in NAFLD with the development of peripheral insulin resistance and impaired pancreatic insulin secretion [103]. As such, while the prevalence of NAFLD in patients with T2DM ranges from 30 to 87% [104,105], NAFLD may also pre-date the development of T2DM in some patients [106]. 

Additional features of metabolic dysfunction, including dyslipidaemia, hypertension, raised high sensitivity C-reactive protein, and advanced atherosclerosis, are over-represented in patients with NAFLD [107,108]. While some of these findings may be due to shared pathophysiologic pathways, some are likely pathogenic. For example, one study has linked NAFLD with increased vascular resistance, higher serum aldosterone, and sympathetic nervous system activation—all potential contributors to hypertension [109]. Others are likely the result of a shared metabolic pathway—some data suggest that there is reduced HDL cholesterol efflux capacity in NAFLD, independently associated with atherosclerosis [110]. This combination of over-expressed pathologic metabolic abnormalities in NAFLD, particularly T2DM and/or obesity, are also associated with an increased risk of liver fibrosis and incident severe liver disease [111]. 

### 2.7. Gut-Liver Axis

The gut microbiome and the gut–liver axis are also implicated in the development and progression of NAFLD [112,113]. While there are a variety of microbiome signatures implicated in NAFLD/NASH, the data are relatively immature, and firm conclusions are necessarily difficult due to potential heterogeneity in studied population ages, geographies, and diets, as well as the actual tools used to sequence the microbiome [114]. Different microbiotal signatures may predispose to NAFLD and liver fibrosis through multiple different pathways which are beyond the scope of this review, though are well-described in Aron-Wisnewsky et al. [114] and Gupta et al. [115]. However, one microbiotal constituent consistently implicated is lipopolysaccharide (LPS), a Gram-negative bacteria cell wall component, which has been recently linked to multiple metabolic diseases including atherosclerosis and T2DM as well as NAFLD [113,116]. By far the most significant source of LPS in humans is from our commensal gut microbiota. In health, the intestinal barrier provides a structural and functional impediment to the translocation of gut bacteria and bacteria-derived antigens (such as LPS) into the portal circulation. While a detailed description of LPS-associated proinflammatory pathways is beyond the scope of this paper, disruption of gut barrier function and LPS translocation causes the release of proinflammatory mediators which can co-activate both hepatic stellate cells and Kupffer cells, causing progressive fibrosis [113]. This disruption and translocation can be caused by multiple pathological pathways: (i) small intestinal bacterial overgrowth [117] (causing excess LPS and translocation); (ii) increased intestinal permeability [118], caused by various factors including exogenous [119] and endogenously-produced [120] ethanol exposure, and/or lifestyle [121]; and (iii) high fat diets [122,123]; all can alter the composition of the intestinal microbiome and/or intestinal permeability. 

#### 2.7.1. Case Identification for NAFLD and NASH

While biopsy remains the gold standard for identifying NAFLD, it is imperfect; there are inherent safety risks [124] (pain, infection, bleeding), liver tissue heterogeneity (leading to sampling error(s) with mis-classification) [125,126], histopathological reporting accuracy [127], and the impracticality of acquiring liver biopsies in all patients suspected of NAFLD. As such, various other surrogate tests are often used in clinical and public health research settings (summarised in Table 2). 

Hepatic steatosis can be identified with reasonable accuracy using ultrasonography [128,129], computed tomography [129,130], and magnetic resonance spectroscopy (MRS) or magnetic resonance imaging protein density fat fraction (MRI-PDFF) [130]. While all these techniques have limitations in terms of sensitivity, their use as a surrogate for liver biopsy is commonplace and appropriate. A recent development in non-invasive assessment is liver stiffness measurement by vibration controlled transient elastography (LSM-VCTE) with an associated controlled attenuation parameter (CAP), allowing for both the identification of hepatic steatosis [131,132] via CAP as well as concurrent assessment of potential fibrosis/cirrhosis [133]. 

Due to time and cost constraints, it is impractical to use liver biopsy, MRI-PDFF, or other radiological assessments on a public health level to determine hepatic steatosis. Therefore, other tools have been developed with greater applicability that combine clinical and biochemical parameters to calculate scores to rule in or out hepatic steatosis (e.g., fatty liver index [134], hepatic steatosis index [135], Framingham steatosis index [136], and Dallas steatosis index [137]). These are accurate and convenient enough for epidemiological modelling and finding associations between conditions in large datasets, though are not yet part of routine clinical practice for identifying individual NAFLD patients. Additionally, while these tests have been externally validated, results may vary by the ethnicity and/or comorbidities of individuals. It is also important to note that the non-invasive identification of NASH is much more difficult than NAFLD. A recent meta-analysis has shown that we lack adequate evidence to differentiate between NAFLD and NASH in clinical practice [138]. Many early composite scores relied on inflammatory biomarkers not collected in routine practice and often lacked the requisite accuracy to alter management [139]. There are emerging novel tests such as the NIS4 [140] which may find a role in risk stratification in the future, though further validation and cost-efficacy assessment will be required.

**Table 2 nutrients-15-00687-t002:** Methods to identify NAFLD.

Surrogate Test for NAFLD Case-ID	Test Components	Sensitivity	Specificity
*Radiology*			
Liver Ultrasound [128,129]		84.8%	93.6%
*Nb*. More sensitive with increasing degrees of steatosis	–		
Liver Computed Tomography [130,141]		72.7–82%	91.3–100%
*Nb*. More sensitive with increasing degrees of steatosis	–		
MRI-PDFF [142]	–	93%	94%
FibroScan CAP [131,132]			
M Probe		79%	74%
*Nb. Cut-off of 294 dB/m can be used to differentiate no steatosis from any degree of steatosis* [132]	–		
XL Probe			
*Nb. Cut-off of 297 dB/m can be used to differentiate no steatosis from any degree of steatosis* [132]		79.8%	73.5%
*Composite Scores*			
Fatty Liver Index [134,143]	BMI, waist circumference, triglycerides, γ-glutamyltransferase		
FLI < 30 (used to rule out NAFLD)		81–87%	64–65%
FLI ≥ 60 (used to rule in NAFLD)		44–61%	86–90%
Framingham Steatosis Index [136,144]	Age, sex, BMI, triglycerides, hypertension, diabetes mellitus, alanine aminotransferase (ALT), aspartate aminotransferase (AST)		
FSI ≥ 23 (used to rule in/out NAFLD)		52–79%	71–80%
Dallas Steatosis Index [137]	Age, sex, BMI, triglycerides, hypertension, diabetes mellitus, ALT, ethnicity, glucose		
<−1.4 risk (low risk of NAFLD)		86%	59%
≥0% risk (high risk of NAFLD)		51%	90%
Hepatic Steatosis Index [135,144]	Sex, BMI, diabetes mellitus, ALT, AST		
HSI < 30 (used to rule out NAFLD)		87.8–93.1%	25–40%
HSI ≥ 36 (used to rule in NAFLD)		25–46%	79–93.1%
NAFLD Liver Fat Score [144,145]	Diabetes mellitus, metabolic syndrome, ALT, AST, fasting insulin		
NAFLD-LFS < −0.640 (used to rule in/out NAFLD)		48.1–86%	71–83.4%

#### 2.7.2. Differential Diagnoses for Hepatic Steatosis

As outlined above, NAFLD requires the exclusion of alternative or secondary causes of steatosis, the most common of which is ethanol [18]. Ethanol’s most direct steatogenic effects are via inhibition of fatty acid oxidation in combination with increased fatty acid synthesis [146]. Additionally, ethanol increases hepatic uptake of fatty acids [147] and impairs a concurrent homeostatic increase in hepatic VLDL synthesis [147]. Ethanol can also alter intestinal barrier function, increasing LPS translocation into the portal system [119] and driving steatosis/fibrosis in a manner similar to that seen in NAFLD. Interestingly, advanced alcohol-related liver disease is associated with *PNPLA3* variants [148] that are classically associated with NAFLD. Mechanistically this is plausible—steatosis, irrespective of cause, is likely worsened by the inability to adequately process hepatocyte lipid droplets. 

Other CLDs may also be steatogenic. Hepatitis C (particularly genotype 3) is associated with hepatic steatosis [149]—and steatosis in that setting may worsen outcomes [14]. Preliminary data suggest that hepatitis C can cause dyslipidaemia and insulin resistance [150]. These may pre-date the development of hepatic steatosis, and the combination of these features has been previously called hepatitis C-associated dysmetabolic syndrome. Wilson’s disease is also associated with hepatic steatosis, though the pathophysiology is complex and incompletely understood [151]. 

Other systemic conditions or disturbances of stable enteral nutrition may also cause hepatic steatosis including total parenteral nutrition (TPN) via small intestinal bacterial overgrowth (due to bowel rest), choline deficiency, protein malnutrition (preventing triglyceride excretion in VLDL particles), excess total calories, and potentially the direct inflammatory and oxidative effects of parenterally-delivered fatty acids [152]. Additionally, refeeding in patients with anorexia nervosa can also lead to hepatic steatosis [153]. polycystic ovarian syndrome (PCOS) is also associated with hepatic steatosis [154,155]; this is due to a combination of insulin resistance, overweight/obesity, and hyperandrogenism. 

Some medications are also considered steatogenic or to promote steatohepatitis. Tamoxifen [156], glucocorticoids [157], amiodarone [158], and multiple others [159] are implicated, and by definition, patients with drug-related steatosis cannot be diagnosed as having NAFLD. As such, careful history taking and biochemical assessment are important in making a firm diagnosis of NAFLD. 

#### 2.7.3. Cirrhosis and Cardiovascular Risk Assessment 

Patients with NAFLD are at risk of progressive fibrosis, cirrhosis, and cardiovascular disease. Liver fibrosis, as a precursor to cirrhosis and subsequent liver-related morbidity and mortality, can be identified through biopsy results (showing features of NASH or progressive fibrosis/cirrhosis) [160], non-invasive devices (e.g., LSM-VCTE) [161], proprietary [140,162,163] or experimental serological tests [164,165] (e.g., enhanced liver fibrosis test [ELF]), or validated non-invasive scores [162] such as FIB-4 [166] and NAFLD fibrosis score [167] (Table 3). Importantly, work has validated the use of many of these scores, particularly LSM-VCTE and FIB-4, to assist with risk stratifying and predicting liver-related outcomes [168]. 

However, when considering these fibrosis assessment tools, many in use were not developed for cardiovascular risk assessment (e.g., FIB-4, LSM-VCTE). Interestingly though, subsequent work has shown an association between worsening liver fibrosis and cardiovascular outcomes [161,169]—as such, these tools may contribute to cardiovascular risk assessment for individuals with NAFLD. Further, other tests (e.g., QRISK3 [170], Framingham risk estimator [171]), while not created specifically for individuals with NAFLD, may still be used to estimate cardiovascular risk in patients with NAFLD. It remains unknown whether those risk scores are accurate for individuals with NAFLD and advanced fibrosis. 

Due to the differential accuracy and cost when utilising these tests, two tier systems have been developed to assist with screening and risk stratifying patients with hepatic steatosis by fibrosis stage in primary care. Generally, these involve the use of an accessible low-cost test (e.g., FIB-4) followed by a second assessment in the case of an indeterminate result. Multiple groups have studied different iterations, including using FIB-4 and ELF [172] or FIB-4 and LSM-VCTE [133,173]. Modelling has shown these two-tier assessments are cost-effective and improve resource utilisation [174,175], though further prospective work in different populations, different countries, and with different funding models will be important.

**Table 3 nutrients-15-00687-t003:** Methods to identify liver fibrosis.

Risk Assessment in NAFLD	Test Components	Sensitivity	Specificity
*Radiology—Advanced Fibrosis*			
Magnetic Resonance Elastography [176]	–	83%	89%
LSM-VCTE [133]			
*Nb*. Different literature uses different cut-offs			
Cut-offs:			
LSM < 7.4 kPa (used to rule out advanced fibrosis)	–	90%	60%
LSM ≥ 12.1 kPa (used to rule in advanced fibrosis)		55%	90%
*Composite Scores—Advanced Fibrosis*			
FIB-4 [133,166]	ALT, AST, platelets, age		
*Nb*. Different literature uses different cut-offs			
Cut-offs:			
FIB-4 < 0.88 (used to rule out advanced fibrosis)		90%	39%
FIB-4 < 1.3 (used to rule out advanced fibrosis)		74%	64%
FIB-4 > 2.31 (used to rule in advanced fibrosis)		38%	90%
FIB-4 > 2.67 (used to rule in advanced fibrosis)		30%	94%
FIB-4 (for those ≥ 65 years of age) [177]	ALT, AST, platelets, age		
*Nb*. Different literature uses different cut-offs			
Cut-offs:			
FIB-4 < 1.3 (used to rule out advanced fibrosis)		93%	35%
FIB-4 > 2.0 (used to rule in advanced fibrosis)		77%	70%
NAFLD Fibrosis Score (NFS) [133]	Age, BMI, impaired fasting glucose, T2DM, ALT, AST, platelets, albumin		
*Nb*. Different literature uses different cut-offs			
Cut-offs:			
NFS < −2.55 (used to rule out advanced fibrosis)		90%	36%
NFS > 0.28 (used to rule in advanced fibrosis)		29%	90%
Enhanced Liver Fibrosis Test [163,172]	Type III procollagen peptide, hyaluronic acid, tissue inhibitor of metalloproteinase-1		
Cut-offs:			
ELF < 9.8 (used to rule out advanced fibrosis)		57.5–65%	86–88.9%
ELF > 11.3 (used to rule in advanced fibrosis)		19.5–36%	96–99.1%
FibroTest [178]	α2 -macroglobulin, apolipoprotein A1, haptoglobin, total bilirubin, γ-glutamyltranspeptidase (GGT)		
Cut-offs:			
FibroTest < 0.30 (used to rule out advanced fibrosis)		92%	71%
FibroTest > 0.70 (used to rule in advanced fibrosis)		25%	97%
BARD Score [162,179]	BMI, ALT, AST, T2DM		
Cut-off:			
Score ≥ 2 (used to rule in/out advanced fibrosis)		75.2%	61.6%
FORNS Index [180,181]	Age, platelets, cholesterol, GGT		
*Nb*. Data are limited for NAFLD			
Cut-offs:			
Score ≤ 4.2 (used to rule out advanced fibrosis)		100%	54.8%
Score > 6.9 (can be used to rule in fibrosis)		42.9%	95.7%
FibroScan-AST (FAST) Score [182]	VCTE-LSM, CAP, AST		
*Nb*. Composite of LSM-VCTE and biochemistry			
Cut-offs:			
FAST ≤ 0.35 (used to rule out advanced fibrosis)		89%	64%
FAST ≥ 0.65 (used to rule in advanced fibrosis)		49%	92%

## 3. Outcomes of NAFLD and NASH

### 3.1. NAFLD—Liver Outcomes

The rate of progression from NAFLD through to NASH, fibrosis, and cirrhosis is both unpredictable and heterogenous [183]. Many patients are likely to not progress at all, however, due to the large population-at-risk, NAFLD is still a significant and growing cause of cirrhosis, liver-related morbidity/mortality, and HCC [7,184]. There are multiple risk factors readily assessed in routine practice that contribute to this risk of progression, including age > 50 years, degree of steatosis, T2DM, and/or obesity. Interestingly, and unlike what is seen in many other liver conditions, the development of HCC in NAFLD can occur prior to the development of advanced fibrosis/cirrhosis. Further, NAFLD/NASH-related liver disease is growing rapidly and is now the second most common indication for liver transplant in the U.S.A. [10]; this trend is likely to extend globally in the near future. Importantly, the progression from simple steatosis through to significant fibrosis and cirrhosis not only increases the risk of liver-related outcomes; worsening biopsy-proven or non-invasive fibrosis stages also correlate with all-cause [160,185] and non-liver [186,187,188] morbidity and mortality. 

### 3.2. NAFLD—Cardiovascular Outcomes

While NAFLD is associated with various aspects of metabolic dysregulation (including dysglycaemia and atherogenic dyslipidaemia), several studies have also shown that NAFLD is independently associated with subclinical [189] and clinical [186] cardiovascular disease. As previously described, NAFLD is associated with increased insulin resistance [13,28] which is itself associated with cardiovascular disease [190]. However, while an in-depth review is outside the scope of this article, there are additional potential mechanisms associated with NAFLD, including systemic inflammation and gut dysbiosis or endotoxemia. Systemic inflammation in NAFLD is associated with cardiovascular disease due to consequent endothelial dysfunction and enhanced atheromatous plaque formation [191,192]. The altered lipid metabolism inherent to NAFLD (characterised by increased VLDL and small dense LDL with reduced HDL) is additionally pro-atherogenic [191,192]. Finally, intestinal dysbiosis is a potential pathogenic feature of NAFLD [112,113] as well as cardiovascular disease [191,192], possibly due to a combination of LPS-associated inflammatory effects alongside pathogenic microbiotal metabolites such as trimethylamine-N-oxide [193,194]. 

### 3.3. NAFLD—Diabetes Mellitus and Chronic Kidney Disease

There is a bi-directional relationship between NAFLD and T2DM [195]. The prevalence of NAFLD in T2DM is as high as 87.1% in some series [104]—however, in patients with NAFLD and without T2DM, rates of insulin resistance and the subsequent development of T2DM are elevated compared with healthy controls. Similarly, patients with NAFLD are at markedly increased risk of developing incident chronic kidney disease (CKD) when compared with healthy controls [187,196]. It is likely that similar pathophysiologic processes associated with CVD in NAFLD drive the association with CKD, including features of the metabolic syndrome and intestinal dysbiosis [196]. 

### 3.4. NAFLD—Obstructive Sleep Apnoea

There is a possible bi-directional relationship between NAFLD and OSA. While OSA is associated with obesity [197] (which is, in turn, associated with NAFLD), the prevalence of hepatic steatosis in patients with OSA may be as high as 75% [198]. Further, OSA may also be associated with high rates of non-invasively assessed liver fibrosis [198]. While many of these findings are primarily driven by BMI, OSA remains independently associated with NAFLD, and it is reasonable to recommend that patients with NAFLD at risk of OSA be appropriately screened and referred. 

### 3.5. Impact of Steatosis on Other Liver Diseases

NAFLD, by definition, cannot co-occur with other liver diseases. In spite of this, however, the interplay between simple steatosis or histopathological features of NASH and other liver diseases is an area of active research. Some of this work is being performed using the new MAFLD nomenclature. Recent work has shown that patients with MAFLD and chronic hepatitis C may have more significant fibrosis and increased histological inflammation than those with chronic hepatitis C without MAFLD [199]. In CHB, one study has shown an association between CHB-MAFLD having reduced transplant-free or HCC-free survival than CHB alone [200], and another has shown that hepatic steatosis improves rates of hepatitis B seroclearance in treatment naïve patients—but still leads to worsening fibrosis stage [201]. While the interplay between steatosis and other liver diseases is a topic of further research, these preliminary data support the role of assessing for hepatic steatosis in patients with chronic viral hepatitis and, by extension, with other liver diseases, and to ensure adequate follow-up of MAFLD subjects irrespective of other liver disease diagnoses. 

## 4. Treatment Options

### 4.1. Lifestyle Modification—Diet 

There are a variety of lifestyle options with which to treat NAFLD—diet and exercise [2,4], coffee consumption [85], and the removal of hepatotoxic drugs (including tobacco [202] and alcohol [203]) are all likely to improve hepatic steatosis/fibrosis. Both weight loss [204,205] and improved dietary composition [204,206] (independent of change in energy intake) can improve steatosis. Weight loss of 7–10% bodyweight is recommended in both European [4] and American [2] guidelines for the management of NAFLD, as it is associated with reductions in histological NASH as well as the potential resolution of steatosis. These guidelines are well supported. A landmark paired biopsy study of hypocaloric diet and exercise showed marked improvements in histological NASH and NAFL activity correlating to the degree of bodyweight lost [205].

Determining whether specific dietary patterns should be recommended has also been a focus of research. When considering weight loss as a specific treatment goal, small studies have shown very low calorie diets (VLCDs) are effective for individuals with NAFLD. One has shown improvement in LSM, LFTs, and cardiovascular risk profile without adverse events [207]. Another small study in patients with T2DM showed improvement in diabetic parameters and hepatic fat content [208]. Dietary macronutrient composition and meal timing has also been examined. One recent randomised trial of a calorie-reduced low-carbohydrate high fat (LCHF) diet vs. intermittent fasting vs. standard of care showed that both dietary interventions improved steatosis and led to reductions in bodyweight [209], though patient satisfaction was better with the intermittent fasting. While hypocaloric LCHF diets may lead to more rapid reductions in intrahepatic lipid content than low fat diets, this is not sustained over the longer term [210,211]. Isocaloric LCHF diets may also reduce intrahepatic lipid content [212], though longer-term follow-up is required. The Mediterranean diet (high in antioxidants, fibre, and UFAs) has also been examined in several trials with conflicting results. Two early studies [206,213] provided support for improvements in ALT, hepatic steatosis, and insulin sensitivity. However, a recent randomised controlled trial showed no clinically significant differences between the Mediterranean diet and a low-fat diet [214]. Despite this, given the Mediterranean diet is shown to improve cardiovascular outcomes [215,216], recommendations to follow a Mediterranean-like diet in NAFLD remain reasonable. 

Coffee intake is recognised as protective in NAFLD, with a recent systematic review showing reduced rates of hepatic fibrosis in individuals with NAFLD who drink coffee [217]. Its role in protecting against simple steatosis is less clear, though a recent dose–response meta-analysis provides some support for high coffee intake (>3 cups per day) being protective against NAFLD development [218]. However, with significant heterogeneity in trials amongst coffee dosing, form (instant, espresso, filter) and caffeine content, definitive conclusions are limited. While recommendations to increase coffee intake are often made [4] (to reduce the risk of developing fibrosis or HCC), further prospective studies in NAFLD are needed to optimise dose and type, and to ensure efficacy. 

### 4.2. Lifestyle Modification—Exercise

Exercise is beneficial for the treatment of NAFLD. In addition to known benefits of exercise in terms of improving insulin resistance [219], dyslipidaemia [220], and hypertension [221], a 2017 review by Hashida et al. [97] highlighted the benefits of both aerobic and anaerobic exercise in improving hepatic steatosis. Both observational and randomised controlled trials almost universally showed improvement in hepatic steatosis, BMI, and ALT levels. Some also showed a modest improvement in hepatic steatosis of 2.65% even without a significant weight change [97]. In practical terms, the median effective exercise duration seen in the Hashida et al. [97] systematic review was 40 min three times per week at 4.8 metabolic equivalents (METs) for aerobic exercise, and a similar duration but lower intensity 45 min three times per week at 3.5 METs for resistance exercise, both for a minimum of 12 weeks. However, the median BMI in these studies was approximately 30 kg/m^2^ and the median age of participants was approximately 49 years. Another recent analysis provides some support to the notion that continuous higher intensity training is more effective than other forms of exercise [222], though it remains unknown how practical intense or long duration exercise would be for those with class III obesity and/or in the elderly. 

### 4.3. Lifestyle Modification—Tobacco and Alcohol

Tobacco smoking—passive and active—contributes to the development of NAFLD [202]. Similarly, active smoking or a smoking history of ≥10 pack years is associated with an increased risk of fibrosis in NAFLD [223]. However, no prospective trials on smoking cessation or on the use of nicotine replacement therapy have been instituted; while smoking cessation is strongly recommended to reduce all-cause and cardiovascular mortality [224], its role in improving liver-specific outcomes is less certain.

Understanding the role of alcohol harm minimisation or cessation in NAFLD is challenging due to the diagnosis of NAFLD mandating no or low-level alcohol intake at baseline. However, some preliminary work has been performed. A recent systematic review notes that even moderate or low-level drinking in NAFLD may worsen liver related outcomes [225]. However, other data suggest that moderate alcohol intake may reduce the risk of CVD-related hospitalisation in NAFLD (with no impact on mortality) [226]. Future trials evaluating the role of alcohol cessation in NAFLD will help to answer these questions. 

### 4.4. Supplementation and Pharmacotherapy

Various therapeutic options have been trialled in NAFLD and NASH to ameliorate disease and reduce histologic severity or cause fibrosis regression. Results to date, however, have been largely disappointing, though multiple trials in late-stage clinical development are on-going (Table 3). Vitamin E (as alpha-tocopherol) was one of the first therapeutic agents to show any potential benefit in NAFLD/NASH—both AASLD and EASL guidelines tentatively recommend its use in biopsy-proven NASH in non-diabetic patients [2,4]. Vitamin E has multiple effects, including as an antioxidant, anti-inflammatory, and apoptosis mediator. Multiple studies have been performed showing some improvements in surrogates of liver damage, steatosis, hepatocellular ballooning, and fibrosis. However, in diabetic patients, the benefits have not been as clearly demonstrated. While one small study showed histological improvement with vitamin E combined with pioglitazone for NAFLD patients with comorbid T2DM, vitamin E alone did not improve any histological markers of NAFLD/NASH apart from steatosis itself [227]. Additionally, vitamin E is not necessarily risk free. A Cochrane review has suggested that there is a trend towards vitamin E supplementation increasing all-cause mortality [228]. While a subsequent analysis of blood tocopherol levels (as a marker of overall intake) has shown no link or slightly reduced risk of mortality [229], given these conflicting results and potential confounding co-occurring phytochemicals when evaluating dietary vitamin E intake, in our experience, many physicians prescribe vitamin E cautiously.

Given that glucose, lipid metabolism, and inflammation have all been implicated in the pathogenesis of NAFLD/NASH, various therapies modulating these pathways have been evaluated. Obeticholic acid (OCA) is one such example. OCA, a Farnesoid X receptor (FXR) agonist and semi-synthetic bile acid analogue, has previously been shown to have anti-inflammatory and anti-fibrotic effects [230]. Following encouraging phase 2 trial data [231], two large randomised controlled trials evaluating the efficacy and safety of OCA in non-cirrhotic NAFLD/NASH have been performed [232,233]. Both showed greater improvements in fibrosis stage with OCA compared with placebo, though side-effects were common, including pruritus and worsening dyslipidaemia. In light of this finding and the need to refer OCA patients to primary care for lipid-lowering therapy in the 2015 RCT [232], a study comparing OCA plus low-dose statin showed that the OCA-induced dyslipidaemia could be successfully ameliorated. Longer term studies evaluating the risk/benefit of combination therapy on clinical liver and cardiovascular outcomes would be useful, however, and multiple OCA trials are on-going (Table 3). 

Due to the metabolic dysregulation and shared pathways linking T2DM and NAFLD/NASH, T2DM pharmaceuticals have also been tried, including pioglitazone and metformin. Pioglitazone, a thiazolidinedione, has been trialled both alone and in combination, for NAFLD/NASH [234,235,236,237]. Thiazolidinediones stimulate PPAR-γ, increasing GLUT1 and GLUT4 levels while concurrently lowering FFAs [238], improving insulin sensitivity, and decreasing hepatic gluconeogenesis, potentially improving histopathological features of NASH [235,237]. Pioglitazone’s benefits have been seen in both a non-diabetic cohort [235] and a T2DM cohort [234,237]. However, in the landmark PIVENS trial which compared pioglitazone, vitamin E, and placebo, pioglitazone showed a numerical improvement in histological response which did not reach the a priori significance level of *p* < 0.025 [236]. Further, concerns around the long-term effects of pioglitazone (weight gain, fracture risk, exacerbating congestive cardiac failure) have likely limited its uptake and use [239]. Given this, newer pan-PPAR agonists such as Lanifibranor are currently being studied (Table 3) but are not yet recommended or approved for routine use. Metformin, a biguanide, has a multitude of incompletely understood actions which sum to reduced hepatic gluconeogenesis and improved peripheral tissue insulin sensitivity [240]. While small trials of metformin have shown some promising results in improving LFTs and BMI, histological findings have been consistently negative [241]. Given these data, pioglitazone has only limited endorsement in the AASLD and EASL NAFLD guidelines, specifically for biopsy proven NASH, and metformin is not currently recommended. 

Newer studies have evaluated more modern T2DM therapies such as sodium–glucose co-transporter 2 (SGLT-2) inhibitors and glucagon-like peptide-1 receptor agonists (GLP-1RAs). SGLT-2 inhibitors function to increase glucosuria by impairing the nephron’s ability to resorb glucose lost through glomerular filtration [242]. Further, they may promote ketogenesis, cause natriuresis, and modulate the renin–aldosterone–angiotensin system; through these, they have anti-obesity, anti-atherosclerotic, and hepato-protective effects in vitro [242]. These findings have been confirmed in patients with T2DM and NAFLD/NASH—small studies have shown the administration of SGLT-2 inhibitors improves ALT, hepatic steatosis, and possibly, fibrosis [243]. GLP-1RAs have also been primarily studied in individuals with T2DM. Early clinical work has shown improvement in steatosis, hepatocyte ballooning, and other histological features of NASH, without significant changes in fibrosis [244]. Further, GLP-1RAs are often used off-label to promote weight loss, even in individuals without T2DM [245,246]. However, similar to SGLT-2 inhibitors, minimal work on NAFLD patients without T2DM has been performed, and recommending these medications for the primary purpose of treating NAFLD/NASH is premature—though trials are on-going (Table 4).

Tirzepatide is a dual-action GLP-1RA and glucose-dependent insulinotropic polypeptide recently trialled in both T2DM [247] and non-T2DM obese patients [248]. In the first trial, improvements in T2DM control were significantly improved compared with semaglutide (a GLP-1RA) with concurrent improvements in transaminases, fasting glucose, insulin sensitivity, lipid profile, and bodyweight [247]. Similar improvements were seen in the obese non-diabetic group [248], and a substudy of a T2DM tirzepatide trial (SURPASS-3 MRI) [249] showed improvements in liver fat content with tirzepatide compared with insulin. However, no histological assessments for improvement/resolution of NASH or fibrosis have been performed to date. However, given these improvements in metabolic profile and steatosis, a trial of tirzepatide in NAFLD/NASH is on-going (Table 3). 

Statins (HMG-CoA reductase inhibitors), while routinely used in the management of cardiovascular disease [250], have been less popular in their use in NAFLD—even in patients with a guideline-supported indication for statin therapy [251] due to potential concerns of hepatotoxicity. However, multiple studies have shown that initiating statin therapy is safe in patients with elevated transaminases, NAFLD/NASH, and/or other underlying liver diseases [252,253]. As such, various studies evaluating the role of statin therapies in improving outcomes in NAFLD/NASH have been performed with mixed results. While early post-hoc and case-controlled analyses have shown improvements in LFTs and radiographic markers of steatosis in statin users compared with non-users, very few reported histopathological results [252]. More recently, a large, nested case-controlled study based on the FLI and BARD scores suggested that statins may reduce the rate of NAFLD development, as well as potentially protect against fibrosis [254]. Further work regarding the protective role of statins on NAFLD, NASH and/or fibrosis is, therefore, needed. There are other potential benefits of statins in NAFLD also. Preliminary work suggests a possible reduction in the risk of developing HCC [255], as well as potentially ameliorating some of the complications of cirrhosis [256]. These findings require confirmation via additional efficacy and safety studies of statin therapy/chemoprophylaxis.

Aspirin has also been evaluated in the management of NAFLD, both regarding the presence of NASH and in terms of preventing cirrhosis and HCC. A prospective cohort study has shown that aspirin users had a time-dependent reduced risk of biopsy-proven NASH and reduced rates of fibrosis progression [257]. Other cross-sectional work supports this—one study showed an inverse relationship between aspirin use and NAFLD prevalence, with results most marked in the elderly and males [258]. Aspirin has also been evaluated for prevention of HCC in CLD—a recent meta-analysis of observational studies confirmed aspirin use conferred a relative risk reduction of 39% [259]. This only included one study of aspirin in confirmed NAFLD, however. As such the role of aspirin as a therapeutic strategy in NAFLD/NASH has not been evaluated in the setting of a randomised-controlled trial and consequently is not a component of NAFLD management guidelines.

### 4.5. Future Directions in NAFLD

As the link between hepatic steatosis and metabolic dysregulation is further crystallised, the need to prospectively evaluate options for improving liver and cardiovascular outcomes in NAFLD (e.g., aspirin and/or statin therapy as primary prevention) should be considered. Further nuanced risk stratification tools and biomarkers will be important to develop a priori, so that scarce healthcare resources can be directed appropriately to those most at risk from this condition that affects upwards of 30% of the Western adult population. Targeting the most appropriate therapies to those most likely to benefit will be critical, especially as many off-label therapies being studied (e.g., GLP-1 agonists) are expensive. Understanding the implications of co-occurring metabolic disease, hepatic steatosis, and other liver diseases (particularly alcohol-related liver disease) will be important to prognosticate and risk-stratify. Finally, country- and health-service-specific prospective evaluations of different models of care will be important so that each region can cost-effectively and proactively manage this public health threat.

## 5. Conclusions

NAFLD is a large and growing cause of liver-related and cardiovascular mortality and morbidity globally. The global burden of disease continues to grow—new risk assessment tools and pathways to manage NAFLD as a component of dysregulated metabolic processes will be vital in combatting this escalating major public health challenge. While there is a large pipeline of repurposed medications to improve the health of patients with NAFLD, identifying those most likely to benefit—and improving lifestyle factors to reduce the incidence of NAFLD—will be critical in reducing the burden of cardiovascular and chronic liver disease in the future.

## Figures and Tables

**Figure 1 nutrients-15-00687-f001:**
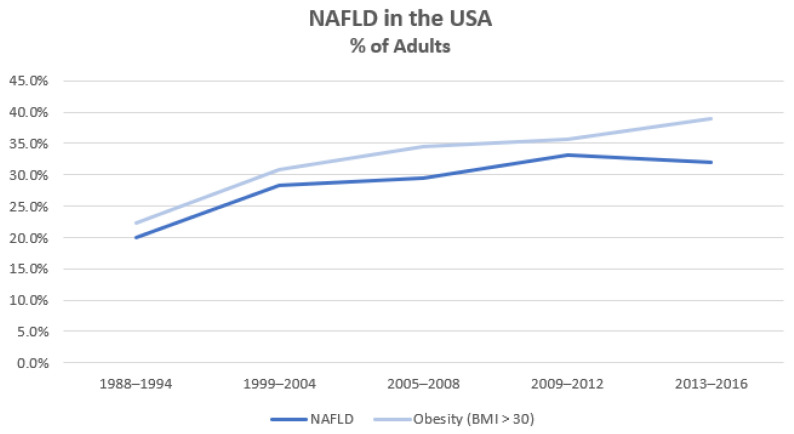
Proportion of U.S.A. adults with NAFLD and/or obesity (adapted from Younossi et al. [3]).

**Table 1 nutrients-15-00687-t001:** Micronutrients and NAFLD/NASH.

Dietary Factor	Epidemiological Association	Proposed Mechanism	Micronutrient Effect on NAFLD Progression
*Vitamins*			
Vitamin E	Poorly defined [66]	Regulates oxidative stress, inflammation, cellular apoptosis [67]	↓ (Sometimes used for treatment)
Vitamin A	Low serum levels in NAFLD/NASH [68]	Regulates de novo lipogenesis and hepatic lipid metabolism [67]	?↓ (Complex interplay [69])
Vitamin C	Low intake in NAFLD/NASH [70]	Regulates oxidative stress, adiponectin	?↓ (Possibly contributes [70])
Vitamin D	(Probably) low in NAFLD/NASH [71,72]	Oxidative stress [72], insulin resistance [67], autophagy [67]	↓
*Minerals*			
Zinc	Mixed data [73,74]	Deficiency may worsen oxidative stress [64] and insulin resistance [75]	?↓ (Animal model [64] and preliminary human [76] data)
Selenium	Mixed data [77]	Antioxidant, apoptosis regulation [77]	?↓ (Reduced fibrosis [78])
Copper	Low hepatic and serum copper in NAFLD/NASH [79]	Oxidative stress, upregulate triglyceride synthesis [64]	↓
Iron	Elevated ferritin associated with NAFLD/NASH severity [80,81]	Hepatic lipid peroxidation [82], impairment of hepatic lipid homeostasis [64]	↑

?: means possible; ↓: means decrease; ↑: means increase.

**Table 4 nutrients-15-00687-t004:** Current late phase pharmaceutical trials in NAFLD (retrieved from ClinicalTrials.gov, accessed on 11 January 2023).

Trial Name	Trial Therapies	Recruiting	ClinicalTrials.gov ID
An Investigator Initiated Prospective, Four Arms Randomized Comparative Study of Efficacy and Safety of Saroglitazar, Vitamin E and Life Style Modification in Patients With Nonalcoholic Fatty Liver Disease (NAFLD)/Non-alcoholic Steatohepatitis (NASH)	Saroglitazar (dual PPAR α/γ agonist) Vitamin E Lifestyle modification	Yes	NCT04193982
Lanifibranor in Patients With Type 2 Diabetes & Nonalcoholic Fatty Liver Disease	Lanifibranor (pan-PPAR agonist) Placebo	Yes	NCT03459079
A Phase 3 Study to Evaluate the Safety and Biomarkers of Resmetirom (MGL-3196) in Non Alcoholic Fatty Liver Disease (NAFLD) Patients (MAESTRO-NAFLD1)	Resmetirom (thyroid hormone receptor β-selective agonist)	No	NCT04197479
A Phase 3 Study to Evaluate Safety and Biomarkers of Resmetirom (MGL-3196) in Patients With Non-alcoholic Fatty Liver Disease (NAFLD), MAESTRO-NAFLD-Open-Label-Extension (MAESTRO-NAFLD-OLE)	Resmetirom (thyroid hormone receptor β-selective agonist)	Yes	NCT04951219
Randomized Global Phase 3 Study to Evaluate the Impact on NASH With Fibrosis of Obeticholic Acid Treatment (REGENERATE)	Obeticholic acid (FXR agonist) Placebo	No	NCT02548351
Study Evaluating the Efficacy and Safety of Obeticholic Acid in Subjects With Compensated Cirrhosis Due to Nonalcoholic Steatohepatitis (REVERSE)	Obeticholic acid (FXR agonist) Placebo	No	NCT03439254
Comparative Study Between Obeticholic Acid Versus Vitamin E in Patients With Non-alcoholic Steatohepatitis	Obeticholic acid (FXR agonist) Vitamin E	No	NCT05573204
Research Study on Whether Semaglutide Works in People With Non-alcoholic Steatohepatitis (NASH) (ESSENCE)	Semaglutide (GLP-1RA) Placebo	Yes	NCT04822181
Study of Semaglutide for Non-Alcoholic Fatty Liver Disease (NAFLD), a Metabolic Syndrome With Insulin Resistance, Increased Hepatic Lipids, and Increased Cardiovascular Disease Risk (The SLIM LIVER Study)	Semaglutide (GLP-1RA)	Yes	NCT04216589
Researching an Effect of GLP-1 Agonist on Liver STeatosis (REALIST) (REALIST)	Dulaglutide (GLP-1RA)	No	NCT03648554
A Study of Tirzepatide (LY3298176) in Participants With Nonalcoholic Steatohepatitis (NASH) (SYNERGY-NASH)	Tirzepatide (dual GLP-1RA/GIP RA) Placebo	Yes	NCT04166773
Dapagliflozin Efficacy and Action in NASH (DEAN)	Dapagliflozin (SGLT-2 inhibitor) Placebo	Yes	NCT03723252
Dapagliflozin in Type 2 Diabetes Mellitus Patients (T2DM) With Nonalcoholic Fatty Liver Disease (NAFLD)	Dapagliflozin (SGLT-2 inhibitor)	Yes	NCT05459701
A Single Center, Randomized, Open Label, Parallel Group, Phase 3 Study to Evaluate the Efficacy of Dapagliflozin in Subjects With Nonalcoholic Fatty Liver Disease	Dapagliflozin (SGLT-2 inhibitor) Placebo	Yes	NCT05308160
Effect of Empagliflozin on Liver Fat in Non-diabetic Patients	Empagliflozin (SGLT-2 inhibitor) Placebo	No	NCT04642261
Efficacy and Safety of Dapagliflozin in Patients With Non-alcoholic Steatohepatitis	Pioglitazone (PPAR-γ agonist) Dapagliflozin (SGLT-2 inhibitor)	Yes	NCT05254626
Canagliflozin on Liver Inflammation Damage in Type 2 Diabetes Patients With Nonalcoholic Fatty Liver Disease	Pioglitazone (PPAR-γ agonist) Canagliflozin (SGLT-2 inhibitor)	No	NCT05422092
Effect of Oral Anti-diabetic Medication on Liver Fat in Subjects With Type II Diabetes and Non-alcoholic Fatty Liver	Pioglitazone (PPAR-γ agonist) Empagliflozin (SGLT-2 inhibitor)	Yes	NCT04976283
Low-Dose Pioglitazone in Patients With NASH (AIM 2)	Pioglitazone (PPAR-γ agonist)	Yes	NCT04501406

## Data Availability

Not applicable.

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
