# Peer review of "Understanding NAFLD: From Case Identification to Interventions, Outcomes, and Future Perspectives"

_nutrients, 2023, doi:10.3390/nu15030687_

Round 1

Reviewer 1 Report (Previous Reviewer 2)

The manuscript has been markedly improved by focusing solely on NAFLD and now provides a well-written, comprehensive review of potential causes, diagnostic criteria used to identify hepatic steatosis and fibrosis, interventions that may or may not improve liver disease and areas future research should focus on. I have only 2 minor comments to address. 

1)    In Section 3 the surrogate tests listed are primarily used to identify hepatic steatosis. Identification of NASH without liver biopsy is much more difficult. A sentence or 2 discussing the ability (or otherwise) to non-invasively detect NASH would improve this section.

2)    In section 7.1 add a sentence or 2 about the effect of low carbohydrate diets of liver fat. 

Author Response

Thank you for your comments and feedback. We have as requested: (1) added text with references in Section 3 regarding the limitations of non-invasive testing for NASH, and (2) included a brief discussion on some of the research into the effects of a low carbohydrate diet on liver fat including a recent RCT of Low Carbohydrate High Fat (LCHF) diet as well as a 2019 meta-analysis & review addressing this issue.

Reviewer 2 Report (New Reviewer)

In the part where hepatic fatty acid metabolism is discussed “2.2. Hepatic Fatty Acid Metabolism” authors should mention, as it is crucial for the pharmacological interventions alongside the pathophysiology of NAFLD mentioning the contribution of different pathways, namely DNL accounting for about 26%, diet 15 % and FFA, derived by AT, ~59%, of the total fatty acids pool (PMID: 35192055). The central role of adipose tissue should be stressed (PMID: 35192055).

Also, excessive FA supply through diet or adipose tissue to the liver does not only enhance TG content (hence NAFL) but also a plethora of other lipids within the liver that might lead to the progression from simple steatosis to more aggressive forms such as NASH, HCC and liver failure (PMID: 34695228). Considering there are almost two decades of literature on the matter, authors should really consider adding something in line with the above suggestion at least mentioning ceramides and diacylglycerols.

In the part where dietary intake is discussed “2.4. Dietary Intake” a line exploring the mechanism by which omega 3 FA promote FA reduction in NAFLD should be added.

“2.5. Physical Activity” Are there any study investigating the role of HIT on NAFLD?.

“2.6. Metabolic Dysfunction” Insulin resistance is a generic term. It would be of interest if authors would mention the different site of IR, namely adipose tissue insulin resistance, hepatic insulin resistance and muscle insulin resistance.

Tables must be aesthetically ameliorated.

Figures graphically showing the main mechanisms of action should be added (especially for the mechanism of action of the drugs).

Author Response

In the part where hepatic fatty acid metabolism is discussed “2.2. Hepatic Fatty Acid Metabolism” authors should mention, as it is crucial for the pharmacological interventions alongside the pathophysiology of NAFLD mentioning the contribution of different pathways, namely DNL accounting for about 26%, diet 15 % and FFA, derived by AT, ~59%, of the total fatty acids pool (PMID: 35192055). The central role of adipose tissue should be stressed (PMID: 35192055). Also, excessive FA supply through diet or adipose tissue to the liver does not only enhance TG content (hence NAFL) but also a plethora of other lipids within the liver that might lead to the progression from simple steatosis to more aggressive forms such as NASH, HCC and liver failure (PMID: 34695228). Considering there are almost two decades of literature on the matter, authors should really consider adding something in line with the above suggestion at least mentioning ceramides and diacylglycerols.
We have expanded on this section to mention ceramides, DAGs, insulin resistance, lipidomics, and the various contributions to intrahepatic triglyceride content including citation of key references above as well as our own recent paper on this topic published in J Hepatology.

In the part where dietary intake is discussed “2.4. Dietary Intake” a line exploring the mechanism by which omega 3 FA promote FA reduction in NAFLD should be added.
We have added the mechanism shown in the cited paper to this sentence.

“2.5. Physical Activity” Are there any study investigating the role of HIT on NAFLD?.
There are multiple definitions of HIT (and HIIT); modified HIIT was examined by Hallsworth et al. and is included in the cited meta-analysis. Another reference has been added to this section to discuss one definition of HIT.

“2.6. Metabolic Dysfunction” Insulin resistance is a generic term. It would be of interest if authors would mention the different site of IR, namely adipose tissue insulin resistance, hepatic insulin resistance and muscle insulin resistance.
We have expanded on this section to better describe the sites and consequences of insulin resistance in these tissues.

Tables must be aesthetically ameliorated.
We aren’t sure what exactly this means. Although the information contained within the Tables is comprehensive in some cases, we believe it is in a format that is quite readable and interpretable particularly given it will be in an online format in this Journal.

Figures graphically showing the main mechanisms of action should be added (especially for the mechanism of action of the drugs).

Unfortunately due to time constraints placed upon us by the Journal for the resubmission, we will be unable to generate a completely new figure nor license and amend a copyright figure for this paper.

Reviewer 3 Report (New Reviewer)

Authors aim to describe several aspects related to NAFLD including  the pathophysiology in the development and progression of NAFLD.

Different tools to identify patients with NAFDL who are at risk of hepatic and cardiovascular related complications  and treatment options for NAFLD are also described in this manuscript.

However the manuscript is difficult to ready mostly due to the long sentences and/or lack of  the focus within the different sections.

I would suggest to rewrite the manuscript  using simple and short sentences and summarize the different sections highlighting the main features.

Author Response

We received this Review only today as we were about to resubmit the revised manuscript. We completely disagree with the reviewer that the manuscript is difficult to read and do not believe a complete rewrite is appropriate. Indeed the two other Reviewers who reviewed the manuscript both believe it is well written. 

Round 2

Reviewer 2 Report (New Reviewer)

Authors have addressed all the required comments in a satisfactory way.

This manuscript is a resubmission of an earlier submission. The following is a list of the peer review reports and author responses from that submission.

Round 1

Reviewer 1 Report

This is an interesting and important review of the literature, this paper explores the differences between MAFLD and NFLD. There are several issues to be clarified.

Q1: In figure 2, I don't know what criteria are used to distinguish NFALD from MAFLD. Not sure what the plus and minus symbols in this figure. How to define these excluded secondary causes in NAFLD and including secondary causes in MAFLD? Figure 2 is hard to read and understand. In reference 18, Eslam et al, the definition and flowchart for the proposed “positive” diagnostic criteria for MAFLD more clear than fig 2 schematic.

Q2.

P.4, 

2. Body? What is the point of what you want to talk?

Q3. p.5 2.4 dietary intake

I suggest that authors can list the effects of macronutrients, micronutrients, and other dietary factors on NAFLD. 

Q4 P.6 line 214

Is there specific microbiome that can affect NAFLD? Here, you only mention the effect of LPS on NAFLD.

Q5.P.8 4.1 Case Identification for NAFLD, NASH and MAFLD & p.9 Table 1 Method to Identify NAFLD, Perhaps it would be more appropriate to move to the head of section 2. To echo the topic is from identification to intervention and outcome.

Q6. P13, 5.2

Please add the effect of time, frequency and type of different exercise on NAFLD or MAFLD in this section.

Q7.

There is a lot of content in the article, discussing NAFLD and MAFLD, but there is little information on MAFLD, and most of them are discussing NAFLD. This literature review needs to be reorganized, otherwise it will be difficult to read.

Author Response

We thank Reviewer 1 for the thoughtful comments and critiques of our manuscript. With regard to the specific queries and suggestions raised, please see below.

Reviewer 1

Comments and Suggestions for Authors

This is an interesting and important review of the literature, this paper explores the differences between MAFLD and NFLD. There are several issues to be clarified.

Q1: In figure 2, I don't know what criteria are used to distinguish NFALD from MAFLD. Not sure what the plus and minus symbols in this figure. How to define these excluded secondary causes in NAFLD and including secondary causes in MAFLD? Figure 2 is hard to read and understand. In reference 18, Eslam et al, the definition and flowchart for the proposed “positive” diagnostic criteria for MAFLD more clear than fig 2 schematic. We believe it’s important to show the difference between NAFLD and MAFLD in one figure, rather than the schematic used to specifically define MAFLD (by Eslam et al.). As such, we have amended the figure to highlight the “positive” vs “negative” criteria, showing that hepatic steatosis is the underlying diagnosis/pathology leading to moving through either definition/paradigm. We have revised this Figure to better illustrate this point.

Q2.

P.4, 

  1. Body? What is the point of what you want to talk? Thank you – we have added further content in the introduction to more clearly explain the purpose of this review and explicitly state what it is not meant to contain.

Q3. p.5 2.4 dietary intake

I suggest that authors can list the effects of macronutrients, micronutrients, and other dietary factors on NAFLD. Thank you – we have added a table detailing the potential influence of key dietary vitamins and minerals on NAFLD/NASH progression. Various other macronutrient and “sub-macronutrient” components are detailed in the text (e.g., saturated vs unsaturated fatty acids).   

Q4 P.6 line 214

Is there specific microbiome that can affect NAFLD? Here, you only mention the effect of LPS on NAFLD. We had focused on LPS as the most clearly described pathobiological mechanism influence NAFLD/NASH and Fibrosis, but for clarity have also added two references for further reading on the topic (as the microbiome in MAFLD/NAFLD/NASH is worthy of a review in itself!)

Q5.P.8 4.1 Case Identification for NAFLD, NASH and MAFLD & p.9 Table 1 Method to Identify NAFLD, Perhaps it would be more appropriate to move to the head of section 2. To echo the topic is from identification to intervention and outcome. We agree – thank you. The sections have been switched to improve readability.

Q6. P13, 5.2

Please add the effect of time, frequency and type of different exercise on NAFLD or MAFLD in this section. We have expanded that section to include the median recommended aerobic and anaerobic exercise by duration, frequency, and intensity, based on the Hashida et al. systematic review.

Q7.

There is a lot of content in the article, discussing NAFLD and MAFLD, but there is little information on MAFLD, and most of them are discussing NAFLD. This literature review needs to be reorganized, otherwise it will be difficult to read. Thank you for your comments and suggestions on improving readability. As you know, the MAFLD field is still relatively immature and growing. There are no comprehensive reviews on the pathophysiology of MAFLD that we are aware of, likely due to both it’s newness as well as the heterogeneity in steatogenesis amongst MAFLD + co-diagnoses (alcohol, Genotype 3 Hepatitis C, etc.). Throughout this review we have aimed to summarise up to date NAFLD information and discuss how expanding diagnostic criteria to allow dual diagnoses may influence how we think about steatogenesis in the future. The actual underlying steatogenic pathways in MAFLD vs NAFLD as single diagnoses without steatogenic contributors 

Reviewer 2 Report

Clayton-Chubb and colleagues review whether non-alcoholic fatty liver disease (NAFLD) or metabolic-dysfunction associated fatty liver disease (MAFLD) should be used when classifying and studying liver disease. Overall the manuscript is well written but reads as a review of NAFLD with little focus on comparing NAFLD to MAFLD. For example, there are large sections devoted to the pathophysiology of NAFLD and methods used to identify this disease with little to no comparison of how this compares to MAFLD. It would be helpful to markedly reduce these sections or place them in the context of MAFLD at all points. A thorough review of the pros and cons of switching from using NAFLD to MAFLD is also missing. For example, is it helpful to group liver disease caused by alcohol intake, viral disease and genetics with obesity and insulin resistance-related liver disease? There may be an increase in mortality rate if MAFLD is used compared to NAFLD but the pathophysiology may very different for people defined as having MAFLD and treatments will differ accordingly. People with familial Hypobetalipoproteinemia will have elevated liver fat for a very different reason to those with increased alcohol intake and hepatitis C. I recommend a substantial rewrite of the manuscript to compare NAFLD to MAFLD in all sections of the manuscript.

Author Response

We thank Reviewer 2 for the thoughtful comments and critiques of our manuscript. With regard to the specific queries and suggestions raised, please see below.

Reviewer 2

Comments and Suggestions for Author: Clayton-Chubb and colleagues review whether non-alcoholic fatty liver disease (NAFLD) or metabolic-dysfunction associated fatty liver disease (MAFLD) should be used when classifying and studying liver disease. Overall the manuscript is well written but reads as a review of NAFLD with little focus on comparing NAFLD to MAFLD. For example, there are large sections devoted to the pathophysiology of NAFLD and methods used to identify this disease with little to no comparison of how this compares to MAFLD. It would be helpful to markedly reduce these sections or place them in the context of MAFLD at all points. A thorough review of the pros and cons of switching from using NAFLD to MAFLD is also missing. For example, is it helpful to group liver disease caused by alcohol intake, viral disease and genetics with obesity and insulin resistance-related liver disease? There may be an increase in mortality rate if MAFLD is used compared to NAFLD but the pathophysiology may very different for people defined as having MAFLD and treatments will differ accordingly. People with familial Hypobetalipoproteinemia will have elevated liver fat for a very different reason to those with increased alcohol intake and hepatitis C. I recommend a substantial rewrite of the manuscript to compare NAFLD to MAFLD in all sections of the manuscript. Thank you for your comments. As can be seen, we’ve made substantive changes to the manuscript based on the comments from reviewer 1 (see above) to both (a) improve the readability of the manuscript and (b) clarify the purpose of this paper. It is important to highlight that the purpose of this review is not to critically appraise the change from NAFLD to MAFLD in terms of utility or policy, but instead highlight the potential steatogenic influences of allowing dual diagnoses including patient outcomes. Similarly, in terms of the critique of MAFLD vs NAFLD, we share some of these concerns – please see second last paragraph of the introduction. We hope this clarifies the purpose of the manuscript, and also highlights the relative immaturity of the MAFLD literature, which limits the ability of our manuscript to include basic science, clinical outcome, and epidemiological implications of MAFLD to NAFLD in a granular way – if we are successful with this paper we aim to generate further discussion and prompt research in just these areas

Round 2

Reviewer 1 Report

I think it isn't a good time to compare NAFLD and MAFLD, because the studies of MAFLD are quiet few. The classification of MAFLD that just been developed in recent years. I suggested that more research data can be collected in the future to compare these two terms from different aspects. The words font of table or figure should be unified.

Reviewer 2 Report

The review still reads as an overview of NAFLD with the comparison to MAFLD included as an afterthought. There were very few changes to the manuscript in response to my comments. The manuscript could easily be modified to focus solely on NAFLD resulting in a stronger review. If the focus remains on comparing NAFLD and MAFLD a substantial rewrite would be required.